# Kronecker Attention Networks

## Abstract

Attention operators have been applied on both 1-D data like texts and higher-order data such as images and videos. Use of attention operators on high-order data requires flattening of the spatial or spatial-temporal dimensions into a vector, which is assumed to follow a multivariate normal distribution. This not only incurs excessive requirements on computational resources, but also fails to preserve structures in data. In this work, we propose to avoid flattening by developing Kronecker attention operators (KAOs) that operate on high-order tensor data directly. KAOs lead to dramatic reductions in computational resources. Moreover, we analyze KAOs theoretically from a probabilistic perspective and point out that KAOs assume the data follow matrix-variate normal distributions. Experimental results show that KAOs reduce the amount of required computational resources by a factor of hundreds, with larger factors for higher-dimensional and higher-order data. Results also show that networks with KAOs outperform models without attention, while achieving competitive performance as those with original attention operators.

## 1 Introduction

Deep learning networks with attention operators have demonstrated great capabilities of solving challenging problems in various tasks such as computer vision (Xu et al., 2015; Lu et al., 2016), natural language processing (Bahdanau et al., 2015; Vaswani et al., 2017), and network embedding (Veličković et al., 2017). Attention operators are capable of capturing long-range relationships and brings significant performance boosts (Li et al., 2018; Malinowski et al., 2018). The application scenarios of attention operators range from 1-D data like texts to high-order and high-dimensional data such as images and videos. However, attention operators suffer from the excessive usage of computational resources when applied on high-order or high-dimensional data. The memory and computational cost increases dramatically with the increase of input orders and dimensions. This prevents attention operators from being applied in broader scenarios. To address this limitation, some studies focus on reducing spatial sizes of inputs such as down-sampling input data (Wang et al., 2018) or attending selected part of data (Huang et al., 2018). However, such kind of methods inevitably results in information and performance loss.

In this work, we propose novel and efficient attention operators, known as Kronecker attention operators (KAOs), for high-order data, which avoid flattening and operate on high-order data directly. Experimental results show that KAOs are as effective as original attention operators, while dramatically reducing the amount of required computational resources. In particular, we employ KAOs to design a family of efficient modules, leading to our compact deep models known as Kronecker attention networks (KANets). KANets significantly outperform prior compact models on the image classification task, with fewer parameters and less computational cost. We also investigate the above problem from a probabilistic perspective. Specifically, regular attention operators flatten the data and assume the flattened data follow multivariate normal distributions. This assumption not only results in high computational cost and memory usage, but also fails to preserve the spatial or spatial-temporal structures of data. Our KAOs, instead, use matrix-variate normal distributions to model the data, where the Kronecker covariance structure is able to capture relationships among spatial or spatial-temporal dimensions.

## 2 Background and Related Work

In this section, we describe the attention and related non-local operators.

## 2.1 Attention Operator

The inputs to an attention operator include a query matrix $\boldsymbol{Q} = [\mathbf{q}_1, \mathbf{q}_2, \cdots, \mathbf{q}_m] \in \mathbb{R}^{d \times m}$, a key matrix $\boldsymbol{K} = [\mathbf{k}_1, \mathbf{k}_2, \cdots, \mathbf{k}_n] \in \mathbb{R}^{d \times n}$, and a value matrix $\boldsymbol{V} = [\mathbf{v}_1, \mathbf{v}_2, \cdots, \mathbf{v}_n] \in \mathbb{R}^{p \times n}$. The attention operation computes the responses of a query vector $\mathbf{q}_i$ by attending it to all key vectors in $\boldsymbol{K}$ and uses the results to take a weighted sum over value vectors in $\boldsymbol{V}$. The layer-wise forward-propagation operation of an attention operator can be expressed as $\boldsymbol{O} = \boldsymbol{V} \text{softmax}(\boldsymbol{K}^T \boldsymbol{Q})$. Matrix multiplication between $\boldsymbol{K}^T$ and $\boldsymbol{Q}$ results in a coefficient matrix $\boldsymbol{E} = \boldsymbol{K}^T \boldsymbol{Q}$, in which each element $e_{ij}$ is calculated by the inner product between $\mathbf{k}_i^T$ and $\mathbf{q}_j$. This coefficient matrix $\boldsymbol{E}$ computes similarity scores between every query vector $\mathbf{q}_i$, and every key vector $\mathbf{k}_j$ and is normalized by a column-wise softmax operator to make every column sum to 1. The output $\boldsymbol{O} \in \mathbb{R}^{p \times m}$ is obtained by multiplying $\boldsymbol{V}$ with the normalized $\boldsymbol{E}$. In self-attention operators (Vaswani et al., 2017), we have $\boldsymbol{Q} = \boldsymbol{K} = \boldsymbol{V}$. The computational cost in attention operator is $O(m \times n \times (d + p))$. The memory required for storing the intermediate coefficient matrix $\boldsymbol{E}$ is $O(mn)$. If $d = p$ and $m = n$, the time and space complexities become $O(m^2 \times d)$ and $O(m^2)$, respectively. There are several other ways to compute $\boldsymbol{E}$ from $\boldsymbol{Q}$ and $\boldsymbol{K}$, including Gaussian function, dot product, concatenation, and embedded Gaussian function. It has been shown that dot product is the simplest but most effective one (Wang et al., 2018). Therefore, we focus on the dot product similarity function in this work. In practice, we can first perform separate linear transformations on each input matrix, resulting in the following attention operator: $\boldsymbol{O} = \boldsymbol{W}^V \boldsymbol{V} \text{Softmax}((\boldsymbol{W}^K \boldsymbol{K})^T \boldsymbol{W}^Q \boldsymbol{Q})$, where $\boldsymbol{W}^V \in \mathbb{R}^{p' \times p}$, $\boldsymbol{W}^K \in \mathbb{R}^{d' \times d}$, and $\boldsymbol{W}^Q \in \mathbb{R}^{d' \times d}$. For notational simplicity, we omit linear transformations in the following discussion.

## 2.2 Non-Local Operator

Non-local operators (Wang et al., 2018) apply self-attention operators on higher-order data such as images and videos. Taking 2-D data as an example, the input to the non-local operator is a third-order tensor $\mathcal{X} \in \mathbb{R}^{h \times w \times c}$, where $h$, $w$, and $c$ denote the height, width, and number of channels, respectively. The tensor is first converted into a matrix $\boldsymbol{X}_{(3)} \in \mathbb{R}^{c \times hw}$ by unfolding along mode-3 (Kolda & Bader, 2009), as illus-

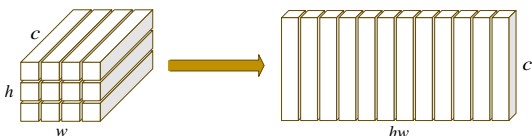

Figure 1: Conversion of a third-order tensor into a matrix by unfolding along mode-3. In this example, a $h \times w \times c$ tensor is unfolded into a $c \times hw$ matrix.

trated in Figure 1. Then we perform the attention operation by setting $\boldsymbol{Q} = \boldsymbol{K} = \boldsymbol{V} = \boldsymbol{X}_{(3)}$. The output of the attention operator is converted back to a third-order tensor as the final output. One practical challenge of the non-local operator is its excessive usage of computational resources. If $h = w$, the computational cost of a 2-D non-local operator is $O(h^4 \times c)$. The memory used to store the coefficient matrix incurs $O(h^4)$ space complexity. The time and space complexities are prohibitively high for high-dimensional and high-order data.

# 3 Kronecker Attention Networks

In this section, we describe our proposed Kronecker attention operators, which are efficient and effective attention operators on high-order data.

## 3.1 Kronecker Attention Operators

We describe the Kronecker attention operators (KAO) in the context of self-attention on 2-D data, but they can be easily generalized to generic attentions. In this case, the input to the $\ell$th layer is a third-order tensor $\mathcal{X}^{(\ell)} \in \mathbb{R}^{h \times w \times c}$. We propose to use horizontal and lateral average matrices to represent original mode-3 unfolding without much information loss. The horizontal average matrix $\boldsymbol{H}$ and the lateral average matrix $\boldsymbol{L}$ are computed as

$$\boldsymbol{H} = \frac{1}{h} \sum_{i=1}^{h} \boldsymbol{X}_{i::}^{(\ell)} \in \mathbb{R}^{w \times c}, \qquad \boldsymbol{L} = \frac{1}{w} \sum_{j=1}^{w} \boldsymbol{X}_{:j:}^{(\ell)} \in \mathbb{R}^{h \times c}, \tag{1}$$

where $\boldsymbol{X}_{i::}^{(\ell)}$ and $\boldsymbol{X}_{:j:}^{(\ell)}$ are the horizontal and lateral slices (Kolda & Bader, 2009) of tensor $\mathcal{X}^{(\ell)}$, respectively. We then form a matrix $\boldsymbol{C}$ by juxtaposing $\boldsymbol{H}^T$ and $\boldsymbol{L}^T$ as $\boldsymbol{C} = [\boldsymbol{H}^T, \boldsymbol{L}^T] \in \mathbb{R}^{c \times (h+w)}$.

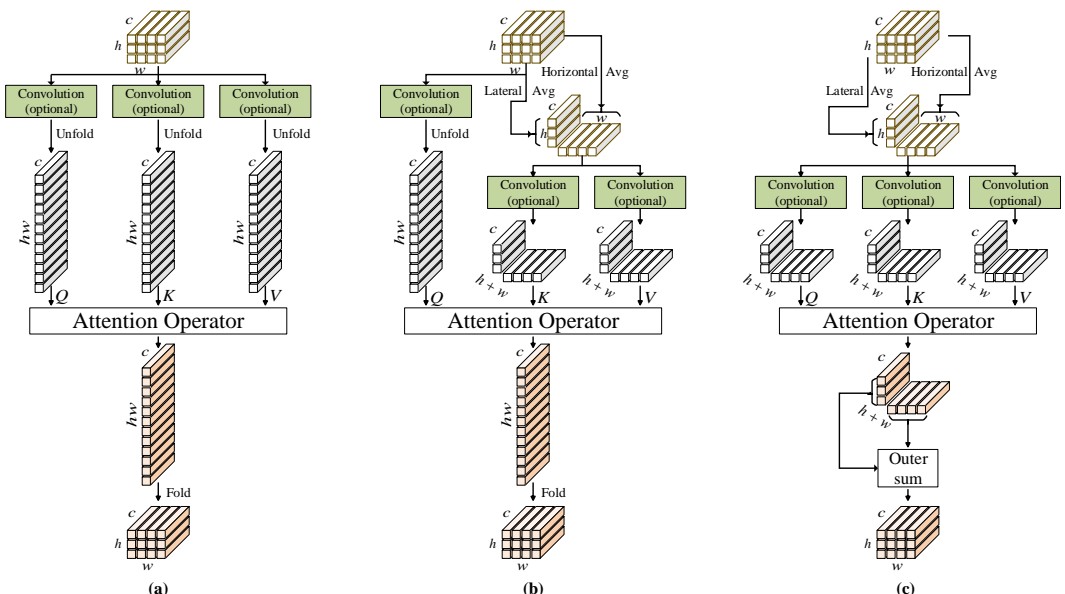

Figure 2: Illustrations of regular attention operator (a), $\text{KAO}_{KV}$ (b) and $\text{KAO}_{QKV}$ (c) on 2-D data. In the regular attention operator (a), the input tensor is unfolded into a mode-3 matrix and fed into the attention operator. The output of the attention operator is folded back to a tensor as the final output. In $\text{KAO}_{KV}$ (b), we juxtapose the horizontal and lateral average matrices derived from the input tensor as the key and value matrices. We keep the mode-3 unfolding of input tensor as the query matrix. In $\text{KAO}_{QKV}$ (c), all three input matrices use the juxtaposition of two average matrices. In contrast to $\text{KAO}_{KV}$, we use an outer-sum operation to generate the output third-order tensor.

Based on the horizontal and lateral average matrices contained in $C$, we propose two Kronecker attention operators (KAOs), *i.e.*, $\text{KAO}_{KV}$ and $\text{KAO}_{QKV}$. In $\text{KAO}_{KV}$ as shown in Figure 2 (b), we use $X_{(3)}^{(\ell)}$ as the query matrix and $C$ as the key and value matrices as

$$O = \text{attn}(X_{(3)}^{(\ell)}, C, C) \in \mathbb{R}^{c \times hw}. \tag{2}$$

Note that the number of columns in $O$ depends on the number of query vectors. Thus, we obtain $hw$ output vectors from the attention operation. Similar to the regular attention operator, $O$ is folded back to a third-order tensor $\mathcal{Y}^{(\ell)} \in \mathbb{R}^{h \times w \times c}$ by considering the column vectors in $O$ as mode-3 fibers of $\mathcal{Y}^{(\ell)}$. $\text{KAO}_{KV}$ uses $\mathcal{Y}^{(\ell)}$ as the output of layer $\ell$. If $h = w$, the time and space complexities of $\text{KAO}_{KV}$ are $O(hw \times c \times (h + w)) = O(h^3 \times c)$ and $O(hw \times (h + w)) = O(h^3)$, respectively. Compared to the original local operator on 2-D data, $\text{KAO}_{KV}$ reduces time and space complexities by a factor of $h$.

In order to reduce the time and space complexities further, we propose another operator known as $\text{KAO}_{QKV}$. In $\text{KAO}_{QKV}$ as shown in Figure 2(c), we use $C$ as the query, key, and value matrices as

$$[\underbrace{\tilde{H}}_{h}, \underbrace{\tilde{L}}_{w}] = O = \text{attn}(C, C, C) \in \mathbb{R}^{c \times (h+w)}. \tag{3}$$

The final output tensor $\mathcal{Y}^{(\ell)} \in \mathbb{R}^{h \times w \times c}$ is obtained as $Y_{::i}^{(\ell)} = \tilde{H}_{i:}^T \oplus \tilde{L}_{i:}^T$, where $\tilde{H}_{i:}$ and $\tilde{L}_{i:}$ are the $i$th rows of the corresponding matrices. That is, the $i$th frontal slice of $\mathcal{Y}^{(\ell)}$ is obtained by computing the outer sum of the $i$th rows of $\tilde{H}$ and $\tilde{L}$. If $h = w$, the time and space complexities of $\text{KAO}_{QKV}$ are $O((h + w) \times c \times (h + w)) = O(h^2 \times c)$ and $O((h + w) \times (h + w)) = O(h^2)$, respectively. Thus, the time and space complexities have been reduced by a factor of $h^2$ as compared to the original local operator, and by a factor of $h$ as compared to $\text{KAO}_{KV}$. Note that we do not consider linear transformations in our description, but they can be applied to all three input matrices in $\text{KAO}_{KV}$ and $\text{KAO}_{QKV}$ as shown in Figure 2.

### 3.2 Kronecker Attention Modules and Networks

Attention models have not been used in compact deep models to date, primarily due to their high computational cost. In this section, we design a family of efficient Kronecker attention modules based on KAOs as illustrated in Figure 3 in the appendix.

**BaseModule:** MobileNetV2 (Sandler et al., 2018) is mainly composed of bottleneck blocks with inverted residuals. Each bottleneck block consists of three convolutional layers; those are, $1 \times 1$ convolutional layer, $3 \times 3$ depth-wise convolutional layer, and another $1 \times 1$ convolutional layer. Suppose the expansion factor is $r$ and stride is $s$. Given input $\mathcal{X}^{(\ell)} \in \mathbb{R}^{h \times w \times c}$ for the $\ell$th block, the first $1 \times 1$ convolutional layer outputs $rc$ feature maps $\tilde{\mathcal{X}}^{(\ell)} \in \mathbb{R}^{h \times w \times rc}$. The depth-wise convolutional layer uses a stride of $s$ and outputs $rc$ feature maps $\bar{\mathcal{X}}^{(\ell)} \in \mathbb{R}^{\frac{h}{s} \times \frac{w}{s} \times rc}$. The last $1 \times 1$ convolutional layer produces $d$ feature maps $\mathcal{Y}^{(\ell)} \in \mathbb{R}^{\frac{h}{s} \times \frac{w}{s} \times d}$. When $s = 1$ and $c = d$, a skip connection is added between $\mathcal{X}^{(\ell)}$ and $\mathcal{Y}^{(\ell)}$.

**BaseSkipModule:** To facilitate feature reuse and gradient back-propagation in deep models, we improve the BaseModule by adding a skip connection. Given input $\mathcal{X}^{(\ell)}$, we use an expansion factor of $r - 1$ for the first $1 \times 1$ convolutional layer, instead of $r$ as in BaseModule. We then concatenate the output with the original input, resulting in $\tilde{\mathcal{X}}^{(\ell)} \in \mathbb{R}^{h \times w \times rc}$. The other parts of the BaseSkipModule are the same as those of the BaseModule. Compared to the BaseModule, the BaseSkipModule reduces the number of parameters by $c \times c$ and computational cost by $h \times w \times c$. It achieves better feature reuse and gradient back-propagation.

**AttnModule:** We propose to add an attention operator into the BaseModule to enable the capture of global features. We reduce the expansion factor of the BaseModule by 1 and add a new parallel path with an attention operator that outputs $c$ feature maps. Concretely, after the depth-wise convolutional layer, the original path outputs $\bar{\mathcal{X}}_a^{(\ell)} \in \mathbb{R}^{\frac{h}{s} \times \frac{w}{s} \times (r-1)c}$. The attention operator, optionally followed by an average pooling of stride $s$ if $s > 1$, produces $\bar{\mathcal{X}}_b^{(\ell)} \in \mathbb{R}^{\frac{h}{s} \times \frac{w}{s} \times c}$. Concatenating them gives $\bar{\mathcal{X}}^{(\ell)} \in \mathbb{R}^{\frac{h}{s} \times \frac{w}{s} \times rc}$. The final $1 \times 1$ convolutional layer remains the same. Within the attention operator, we only apply the linear transformation on the value matrix $\boldsymbol{V}$ to limit the number of parameters and required computational resources. In this module, the original path acts as locality-based feature extractors, while the new path with an attention operator computes global features. This enables the module to incorporate both local and global information. Note that we can use any attention operator in this module, including the regular attention operator and our KAOs.

**AttnSkipModule:** We propose to add an additional skip connection in the AttnModule. This skip connection can always be added unless $s > 1$. The AttnSkipModule has the same amount of parameters and computational cost as the AttnModule.

In the following sections, we perform some theoretical analysis on the proposed methods.

### 3.3 From Multivariate to Matrix-Variate Distributions

We analyze our solutions for attention operators on high-order data from a probabilistic perspective. We take the non-local operator on 2-D data as an example. Formally, consider a self-attention operator with $\boldsymbol{Q} = \boldsymbol{K} = \boldsymbol{V} = \boldsymbol{X}_{(3)}$, where $\boldsymbol{X}_{(3)} \in \mathbb{R}^{c \times hw}$ is the mode-3 unfolding of a third-order input tensor $\mathcal{X} \in \mathbb{R}^{h \times w \times c}$, as illustrated in Figure 1. The $i$th row of $\boldsymbol{X}_{(3)}$ corresponds to $\text{vec}(\boldsymbol{X}_{::i})^T \in \mathbb{R}^{hw}$, where $\boldsymbol{X}_{::i} \in \mathbb{R}^{h \times w}$ denotes the $i$th frontal slice of $\mathcal{X}$ (Kolda & Bader, 2009), and $\text{vec}(\cdot)$ denotes the vectorization of a matrix by concatenating its columns (Gupta & Nagar, 2018).

The frontal slices $\boldsymbol{X}_{::1}, \boldsymbol{X}_{::2}, \ldots, \boldsymbol{X}_{::c} \in \mathbb{R}^{h \times w}$ of $\mathcal{X}$ are usually known as $c$ feature maps. In this view, the mode-3 unfolding is equivalent to the vectorization of each feature map independently. It is worth noting that, in addition to $\text{vec}(\cdot)$, any other operation that transforms each feature map into a vector leads to the same output from the non-local operator, as long as a corresponding reverse operation is performed to fold the output into a tensor. This fact indicates that unfolding of $\mathcal{X}$ in local operators ignores the structural information within each feature map, *i.e.,* the relationships among rows and columns. In addition, such unfolding results in excessive requirements on computational resources, as explained in Section 2.2.

In the following discussions, we focus on one feature map $X \in \{X_{::1}, X_{::2}, \ldots, X_{::c}\}$ by assuming feature maps are conditionally independent of each other, given feature maps of previous layers. This assumption is shared by many deep learning techniques that process each feature map independently, including the unfolding mentioned above, batch normalization, instance normalization (Ulyanov et al., 2016), and pooling operations (LeCun et al., 1998). To view the problem above from a probabilistic perspective (Ioffe & Szegedy, 2015; Ulyanov et al., 2016), the unfolding yields the assumption that $\text{vec}(X)$ follows a multivariate normal distribution as $\text{vec}(X) \sim \mathcal{N}_{hw}(\boldsymbol{\mu}, \boldsymbol{\Omega})$, where $\boldsymbol{\mu} \in \mathbb{R}^{hw}$ and $\boldsymbol{\Omega} \in \mathbb{R}^{hw \times hw}$. Apparently, the multivariate normal distribution does not explicitly model relationships among rows and columns in $X$. To address this limitation, we propose to model $X$ using a matrix-variate normal distribution (Gupta & Nagar, 2018), defined as below.

**Definition 1.** A random matrix $A \in \mathbb{R}^{m \times n}$ is said to follow a matrix-variate normal distribution $\mathcal{MN}_{m \times n}(M, \boldsymbol{\Omega} \otimes \boldsymbol{\Psi})$ with mean matrix $M \in \mathbb{R}^{m \times n}$ and covariance matrix $\boldsymbol{\Omega} \otimes \boldsymbol{\Psi}$, where $\boldsymbol{\Omega} \in \mathbb{R}^{m \times m} > 0$ and $\boldsymbol{\Psi} \in \mathbb{R}^{n \times n} > 0$, if $\text{vec}(A^T) \sim \mathcal{N}_{mn}(\text{vec}(M^T), \boldsymbol{\Omega} \otimes \boldsymbol{\Psi})$. Here, $\otimes$ denotes the Kronecker product (Van Loan, 2000; Graham, 2018).

The matrix-variate normal distribution has separate covariance matrices for rows and columns. They interact through the Kronecker product to produce the covariance matrix for the original distribution. Specifically, for two elements $X_{ij}$ and $X_{i'j'}$ from different rows and columns in $X$, the relationship between $X_{ij}$ and $X_{i'j'}$ is modeled by the interactions between the $i$th and $i'$th rows and the $j$th and $j'$th columns. Therefore, the matrix-variate normal distribution can incorporate relationships among rows and columns.

### 3.4 THE PROPOSED MEAN AND COVARIANCE STRUCTURES

In machine learning, (Kalaitzis et al., 2013) proposed to use the Kronecker sum to form covariance matrices, instead of the Kronecker product. Based on the above observations and studies, we propose to model $X$ as $X \sim \mathcal{MN}_{h \times w}(M, \boldsymbol{\Omega} \oplus \boldsymbol{\Psi})$, where $M \in \mathbb{R}^{h \times w}$, $\boldsymbol{\Omega} \in \mathbb{R}^{h \times h} > 0$, $\boldsymbol{\Psi} \in \mathbb{R}^{w \times w} > 0$, $\oplus$ denotes the Kronecker sum (Kalaitzis et al., 2013), defined as $\boldsymbol{\Omega} \oplus \boldsymbol{\Psi} = \boldsymbol{\Omega} \otimes I_{[w]} + I_{[h]} \otimes \boldsymbol{\Psi}$, and $I_{[n]}$ denotes an $n \times n$ identity matrix. Covariance matrices following the Kronecker sum structure can still capture the relationships among rows and columns (Kalaitzis et al., 2013). It also follows from (Allen & Tibshirani, 2010; Wang et al., 2017) that constraining the mean matrix $M$ allows a more direct modeling of the structural information within a feature map. Following these studies, we assume $X$ follows a variant of the matrix-variate normal distribution as

$$X \sim \mathcal{MN}_{h \times w}(M, \boldsymbol{\Omega} \oplus \boldsymbol{\Psi}), \tag{4}$$

where the mean matrix $M \in \mathbb{R}^{h \times w}$ is restricted to be the outer sum of two vectors, defined as $M = \boldsymbol{\mu} \oplus \boldsymbol{v} = \boldsymbol{\mu} \mathbf{1}_{[w]}^T + \mathbf{1}_{[h]} \boldsymbol{v}^T$, where $\boldsymbol{\mu} \in \mathbb{R}^h$, $\boldsymbol{v} \in \mathbb{R}^w$, and $\mathbf{1}_{[n]}$ denotes an all-one vector of size $n$.

Under this model, the marginal distributions of rows and columns are both multivariate normal (Allen & Tibshirani, 2010). Specifically, the $i$th row vector $X_{i:} \in \mathbb{R}^{1 \times w}$ follows $X_{i:}^T \sim \mathcal{N}_w(\mu_i + \boldsymbol{v}^T, \Omega_{ii} + \boldsymbol{\Psi})$, and the $j$th column vector $X_{:j} \in \mathbb{R}^{h \times 1}$ follows $X_{:j} \sim \mathcal{N}_h(v_i + \boldsymbol{\mu}, \Psi_{ii} + \boldsymbol{\Omega})$. In the following discussion, we assume that $\boldsymbol{\Omega}$ and $\boldsymbol{\Psi}$ are diagonal, implying that any pair of variables in $X$ are uncorrelated. Note that, although the variables in $X$ are independent, their covariance matrix still follows the Kronecker covariance structure, thus capturing the relationships among rows and columns (Allen & Tibshirani, 2010; Wang et al., 2017).

### 3.5 MAIN TECHNICAL RESULTS

Let $\overline{X}_{row} = (\sum_{i=1}^h X_{i:}^T)/h \in \mathbb{R}^w$ and $\overline{X}_{col} = (\sum_{j=1}^w X_{:j})/w \in \mathbb{R}^h$ be the average of row and column vectors, respectively. Under the assumption above, $\overline{X}_{row}$ and $\overline{X}_{col}$ follow multivariate normal distributions as

$$\overline{X}_{row} \sim \mathcal{N}_w(\overline{\boldsymbol{\mu}} + \boldsymbol{v}, \frac{\overline{\boldsymbol{\Omega}} + \boldsymbol{\Psi}}{h}) \qquad (5) \qquad\qquad \overline{X}_{col} \sim \mathcal{N}_h(\overline{\boldsymbol{v}} + \boldsymbol{\mu}, \frac{\overline{\boldsymbol{\Psi}} + \boldsymbol{\Omega}}{w}), \qquad (6)$$

where $\overline{\boldsymbol{\mu}} = (\sum_{i=1}^h \mu_i)/h$, $\overline{\boldsymbol{\Omega}} = (\sum_{i=1}^h \Omega_{ii})/h$, $\overline{\boldsymbol{v}} = (\sum_{j=1}^w v_j)/w$, and $\overline{\boldsymbol{\Psi}} = (\sum_{j=1}^w \Psi_{jj})/w$. Our main technical results can be summarized in the following theorem.

Table 1: Comparisons between the regular attention operator, the regular attention operator with a pooling operation (Wang et al., 2018), and our proposed $\text{KAO}_{KV}$ and $\text{KAO}_{QKV}$ in terms of the number of parameters, MAdd, memory usage, and CPU inference time on data of different sizes. The input sizes are given in the format of "batch size × spatial sizes × number of channels". "Attn" denotes the regular attention operator. "Attn+Pool" denotes the regular attention operator which employs a $2 \times 2$ pooling operation on $K$ and $V$ input matrices to reduce required computational resources.

| Input | Operator | MAdd | Memory | Saving | Time | Speedup |
|---|---|---|---|---|---|---|
| $8 \times 14^2 \times 8$ | Attn | 0.63m | 5.2MB | 0.00% | 5.8ms | 1.0× |
| | Attn+Pool | 0.16m | 1.5MB | 71.65% | 2.0ms | 3.0× |
| | $\text{KAO}_{KV}$ | 0.09m | 0.9MB | 82.03% | 1.7ms | 3.5× |
| | $\text{KAO}_{QKV}$ | **0.01m** | **0.3MB** | **95.06%** | **0.8ms** | **6.8×** |
| $8 \times 28^2 \times 8$ | Attn | 9.88m | 79.9MB | 0.00% | 72.4ms | 1.0× |
| | Attn+Pool | 2.47m | 20.7MB | 74.13% | 20.9ms | 3.5× |
| | $\text{KAO}_{KV}$ | 0.71m | 6.5MB | 91.88% | 7.1ms | 10.1× |
| | $\text{KAO}_{QKV}$ | **0.05m** | **0.9MB** | **98.85%** | **1.7ms** | **40.9×** |
| $8 \times 56^2 \times 8$ | Attn | 157.55m | 1,262.6MB | 0.00% | 1,541.1ms | 1.0× |
| | Attn+Pool | 39.39m | 318.7MB | 74.76% | 396.9ms | 3.9× |
| | $\text{KAO}_{KV}$ | 5.62m | 48.2MB | 96.18% | 49.6ms | 31.1× |
| | $\text{KAO}_{QKV}$ | **0.21m** | **3.4MB** | **99.73%** | **5.1ms** | **305.8×** |

**Theorem 1.** Given the multivariate normal distributions in Eqs. (5) and (6) with diagonal $\mathbf{\Omega}$ and $\mathbf{\Psi}$, if *(a)* $\mathbf{r}_1, \mathbf{r}_2, \ldots, \mathbf{r}_h$ are independent and identically distributed (i.i.d.) random vectors that follow the distribution in Eq. (5), *(b)* $\mathbf{c}_1, \mathbf{c}_2, \ldots, \mathbf{c}_w$ are i.i.d. random vectors that follow the distribution in Eq. (6), *(c)* $\mathbf{r}_1, \mathbf{r}_2, \ldots, \mathbf{r}_h$ and $\mathbf{c}_1, \mathbf{c}_2, \ldots, \mathbf{c}_w$ are independent, we have $\tilde{X} \sim \mathcal{MN}_{h \times w} \left( \tilde{M}, \frac{\overline{\mathbf{\Psi}} + \mathbf{\Omega}}{w} \oplus \frac{\overline{\mathbf{\Omega}} + \mathbf{\Psi}}{h} \right)$, where $\tilde{X} = [\mathbf{r}_1, \mathbf{r}_2, \ldots, \mathbf{r}_h]^T + [\mathbf{c}_1, \mathbf{c}_2, \ldots, \mathbf{c}_w]$, $\tilde{M} = (\boldsymbol{\mu} \oplus \boldsymbol{\upsilon}) + (\overline{\boldsymbol{\mu}} + \overline{\boldsymbol{\upsilon}})$. In particular, if $h = w$, the covariance matrix satisfies $\text{tr} \left( \frac{\overline{\mathbf{\Psi}} + \mathbf{\Omega}}{w} \oplus \frac{\overline{\mathbf{\Omega}} + \mathbf{\Psi}}{h} \right) = \frac{2}{h} \text{tr} \left( \mathbf{\Omega} \oplus \mathbf{\Psi} \right)$, where $\text{tr}(\cdot)$ denotes matrix trace.

The proof of Theorem 1 can be found in the appendix. With certain normalization on $X$, we can have $\overline{\boldsymbol{\mu}} + \overline{\boldsymbol{\upsilon}} = 0$, resulting in $\tilde{M} = \boldsymbol{\mu} \oplus \boldsymbol{\upsilon}$. As the trace of a covariance matrix measures the total variation, Theorem 1 implies that $\tilde{X}$ follows a matrix-variate normal distribution with the same mean and scaled covariance as the distribution of $X$ in Eq. (4). We build KAOs based on this conclusion and the process to obtain $\tilde{X}$ from $X$.

# 4 EXPERIMENTAL STUDIES

In this section, we evaluate our methods and networks on image classification and segmentation tasks.

## 4.1 COMPARISON OF COMPUTATIONAL EFFICIENCY

According to the theoretical analysis in Section 3.1, our KAOs have efficiency advantages over regular attention operators on high-order data, especially for inputs with large spatial sizes. We conduct simulated experiments to evaluate the theoretical results. To reduce the influence of external factors, we build networks composed of a single attention operator, and apply the TensorFlow profile tool (Abadi et al., 2016) to report the multiply-adds (MAdd), required memory, and time consumed on 2-D simulated data. For the simulated input data, we set the batch size and number of channels both to 8, and test three spatial sizes; those are, $56 \times 56$, $28 \times 28$, and $14 \times 14$. The number of output channels is also set to 8. Table 1 summarizes the comparison results. On simulated data of spatial sizes $56 \times 56$, our $\text{KAO}_{KV}$ and $\text{KAO}_{QKV}$ achieve 31.1 and 305.8 times speedup, and 96.18% and 99.73% memory saving compared to the regular attention operator, respectively. Our proposed KAOs show significant improvements over regular attention operators in terms of computational resources, which is consistent with the theoretical analysis. In particular, the amount of improvement increases as the spatial sizes increase. These results show that the proposed KAOs are efficient attention operators on high-dimensional and high-order data.

## 4.2 RESULTS ON IMAGE CLASSIFICATION

With the high efficiency of our KAOs, we have proposed several efficient Kronecker attention modules for compact CNNs in Section 3.2. To further show the effectiveness of KAOs and the modules, we build novel compact CNNs known as Kronecker attention networks (KANets). Following the practices in (Wang et al., 2018), we apply these modules on inputs of spatial sizes $28 \times 28$, $14 \times 14$, and $7 \times 7$. The detailed network architecture is described in Table 6 in the appendix due to space constraint. We compare KANets with other CNNs on the ImageNet ILSVRC 2012 image classification dataset, which serves as the benchmark for compact CNNs (Howard et al., 2017; Zhang et al., 2017; Gao et al., 2018; Sandler et al., 2018). Details of the experimental setups are provided in the appendix.

The comparison results between our KANets and other CNNs in terms of the top-1 accuracy, number of parameters, and MAdd are reported in Table 2. SqueezeNet (Iandola et al., 2016) has the least number of parameters, but uses the most MAdd and does not obtain competitive performance as compared to other compact CNNs. Among compact CNNs, MobileNetV2 (Sandler et al., 2018) is the previous state-of-the-art model, which achieves the best trade-off between effectiveness and efficiency. According to the results, our KANets significantly outperform MobileNetV2 with 0.03 million fewer parameters. Specifically, our $KANet_{KV}$ and $KANet_{QKV}$ outperform MobileNetV2 by margins of 0.9% and 0.8%, respectively. More importantly, our KANets has the least computational cost. These results demonstrate the effectiveness and efficiency of our proposed KAOs.

Table 2: Comparisons between KANets and other CNNs in terms of the top-1 accuracy on the ImageNet validation set, the number of total parameters, and MAdd. We use $KANet_{KV}$ and $KANet_{QKV}$ to denote KANets using $KAO_{KV}$ and $KAO_{QKV}$, respectively.

| Model | Top-1 | Params | MAdd |
|---|---|---|---|
| VGG16 | 0.715 | 128m | 15300m |
| AlexNet | 0.572 | 60m | 720m |
| SqueezeNet | 0.575 | 1.3m | 833m |
| MobileNetV1 | 0.706 | 4.2m | 569m |
| ShuffleNet 1.5x | 0.715 | 3.4m | 292m |
| ChannelNet-v1 | 0.705 | 3.7m | 407m |
| MobileNetV2 | 0.720 | 3.47m | 300m |
| **KANet$_{KV}$** (ours) | **0.729** | 3.44m | 288m |
| **KANet$_{QKV}$** (ours) | 0.728 | 3.44m | **281m** |

The performance of KANets indicates that our proposed methods are promising, since we only make small modifications to the architecture of MobileNetV2 to include KAOs. Compared to modules with the regular convolutional layers only, our proposed modules with KAOs achieve better performance without using excessive computational resources. Thus, our methods can be used widely for designing compact deep models. Next, we show that our proposed KAOs are as effective as regular attention operators.

## 4.3 COMPARISON WITH REGULAR ATTENTION OPERATORS

We perform experiments to compare our proposed KAOs with regular attention operators. We consider the regular attention operator and the one with a pooling operation in (Wang et al., 2018). For the attention operator with pooling operation, the spatial sizes of the key matrix $K$ and value matrix $V$ are reduced by $2 \times 2$ pooling operations to save computation cost. To compare these operators in fair settings, we replace all KAOs in KANets with regular attention operators and regular attention operators with a pooling operation, denoted as AttnNet and AttnNet+Pool, respectively.

Table 3: Comparisons between KANets with regular attention operators (denoted as AttnNet), KANets with regular attention operators with a pooling operation (denoted as AttnNet+Pool) and KANets with KAOs in terms of the top-1 accuracy on the ImageNet validation set, the number of total parameters, and MAdd.

| Model | Top-1 | Params | MAdd |
|---|---|---|---|
| AttnNet | 0.730 | 3.44m | 365m |
| AttnNet+Pool | 0.729 | 3.44m | 300m |
| KANet$_{KV}$ | 0.729 | 3.44m | 288m |
| KANet$_{QKV}$ | 0.728 | 3.44m | **281m** |

The comparison results are summarized in Table 3. Note that all these models have the same number of parameters. We can see that $KANet_{KV}$ and $KANet_{QKV}$ achieve similar performance as AttnNet and AttnNet+Pool with dramatic reductions of computational cost. The results indicate that our proposed KAOs are as effective as regular attention operators while being much more efficient. In addition, our KAOs are better than regular attention operators that uses a pooling operation to increase efficiency in (Wang et al., 2018).

## 4.4 ABLATION STUDIES

To show how our KAOs benefit entire networks in different settings, we conduct ablation studies on MobileNetV2 and KANet$_{KV}$. For MobileNetV2, we replace BaseModules with Attn-Modules as described in Section 3.2, resulting in a new model denoted as MobileNetV2+KAO. On the contrary, based on KANet$_{KV}$, we replace all AttnSkipModules by BaseModules. The resulting model is denoted as KANet w/o KAO.

Table 4 reports the comparison results. By employing KAO$_{KV}$, MobileNetV2+KAO gains a performance boost of 0.6% with fewer parameters than MobileNetV2. On the other hand, KANet$_{KV}$ outperforms KANet w/o KAO by a margin of 0.8%, while KANet w/o KAO has more parameters than KANet$_{KV}$. KANet$_{KV}$ achieves the best performance while costing the least computational resources. The results indicate that our proposed KAOs are effective and efficient, which is independent of specific network architectures.

Table 4: Comparisons among KANet$_{KV}$, MobileNetV2, MobileNetV2 with KAOs$_{KV}$ (MobileNetV2+KAO$_{KV}$), and KANet without KAO (KANet w/o KAO) in terms of the top-1 accuracy on the ImageNet validation set, the number of total parameters, and MAdd.

| Model | Top-1 | Params | MAdd |
|---|---|---|---|
| MobileNetV2 | 0.720 | 3.47m | 300m |
| MobileNetV2+KAO | 0.726 | 3.46m | 298m |
| KANet$_{KV}$ | **0.729** | **3.44m** | **288m** |
| KANet w/o KAO | 0.721 | 3.46m | 298m |

## 4.5 RESULTS ON IMAGE SEGMENTATION

In order to show the efficiency and effectiveness of our KAOs in broader application scenarios, we perform additional experiments on image segmentation tasks using the PASCAL 2012 dataset (Everingham et al., 2010). With the extra annotations provided by (Hariharan et al., 2011), the augmented dataset contains 10,582 training, 1,449 validation, and 1,456 testing images. Each pixel of the images is labeled by one of 21 classes with 20 foreground classes and 1 background class. We re-implement the DeepLabV2 model (Chen et al., 2018) as our baseline. Following (Wang & Ji, 2018), using attention operators as the output layer, instead of atrous spatial pyramid pooling (ASPP), results in a significant performance improvement. In our experiments, we replace ASPP with the regular attention operator and our proposed KAOs, respectively, and compare the results. For all attention operators, linear transformations are applied on $Q$, $K$, and $V$. Details of the experimental setups are provided in the appendix.

Table 5: Comparisons among DeepLabV2, DeepLabV2 with the regular attention operator (DeepLabV2+Attn), DeepLabV2 with KAO$_{KV}$ (DeepLabV2+KAO$_{KV}$), and DeepLabV2 with KAO$_{QKV}$ (DeepLabV2+KAO$_{QKV}$) in terms of the pixel-wise accuracy, and mean IOU on the PASCAL VOC 2012 validation dataset.

| Model | Accuracy | Mean IOU |
|---|---|---|
| DeepLabV2 | 0.944 | 75.1 |
| DeepLabV2+Attn | 0.947 | 76.3 |
| DeepLabV2+KAO$_{KV}$ | 0.946 | 75.9 |
| DeepLabV2+KAO$_{QKV}$ | 0.946 | 75.8 |

Table 5 shows the evaluation results in terms of pixel accuracy and mean intersection over union (IoU) on the PASCAL VOC 2012 validation set. Clearly, models with attention operators outperform the baseline model with ASPP. Compared with the regular attention operator, KAOs result in similar pixel-wise accuracy but slightly lower mean IoU. From the pixel-wise accuracy, results indicate that KAOs are as effective as the regular attention operator. The decrease in mean IoU may be caused by the strong structural assumption behind KAOs. Overall, the experimental results demonstrate the efficiency and effectiveness of our KAOs in broader application scenarios.

## 5 CONCLUSIONS

In this work, we propose Kronecker attention operators to address the practical challenge of applying attention operators on high-order data. We investigate the problem from a probabilistic perspective and use matrix-variate normal distributions with Kronecker covariance structure. Experimental results show that our KAOs reduce the amount of required computational resources by a factor of hundreds, with larger factors for higher-dimensional and higher-order data. We employ KAOs to design a family of efficient modules, leading to our KANets. KANets significantly outperform the previous state-of-the-art compact models on image classification tasks, with fewer parameters and less computational cost.

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

# Appendix

## 1 EXPERIMENTAL SETUP FOR IMAGE CLASSIFICATION

As a common practice on this dataset, we use the same data augmentation scheme in (He et al., 2016). Specifically, during training, we scale each image to $256 \times 256$ and then randomly crop a $224 \times 224$ patch. During inference, the center-cropped patches are used. We train our KANets using the same settings as MobileNetV2 (Sandler et al., 2018) with minor changes. We perform batch normalization (Ioffe & Szegedy, 2015) on the coefficient matrices in KAOs to stabilize the training. All trainable parameters are initialized with the Xavier initialization (Glorot & Bengio, 2010). We use the standard stochastic gradient descent optimizer with a momentum of 0.9 (Sutskever et al., 2013) to train models for 150 epochs in total. The initial learning rate is 0.1 and it decays by 0.1 at the 80th, 105th, and 120th epoch. Dropout (Srivastava et al., 2014) with a keep rate of 0.8 is applied after the global average pooling layer. We use 8 TITAN Xp GPUs and a batch size of 512 for training, which takes about 1.5 days. Since labels of the test dataset are not available, we train our networks on training dataset and report accuracies on the validation dataset.

## 2 EXPERIMENTAL SETUP FOR IMAGE SEGMENTATION

We train all the models with randomly cropped patches of size $321 \times 321$ and a batch size of 8. Data augmentation by randomly scaling the inputs for training is employed. We adopt the "poly" learning rate policy (Liu et al., 2015) with $power = 0.9$, and set the initial learning rate to 0.00025. Following DeepLabV2, we use the ResNet-101 model pre-trained on ImageNet (Deng et al., 2009) and MS-COCO (Lin et al., 2014) for initialization. The models are then trained for 25,000 iterations with a momentum of 0.9 and a weight decay of 0.0005. We perform no post-processing such as conditional random fields and do not use multi-scale inputs due to limited GPU memory. All the models are trained on the training set and evaluated on the validation set.

## 3 ILLUSTRATION OF KRONECKER ATTENTION MODULES

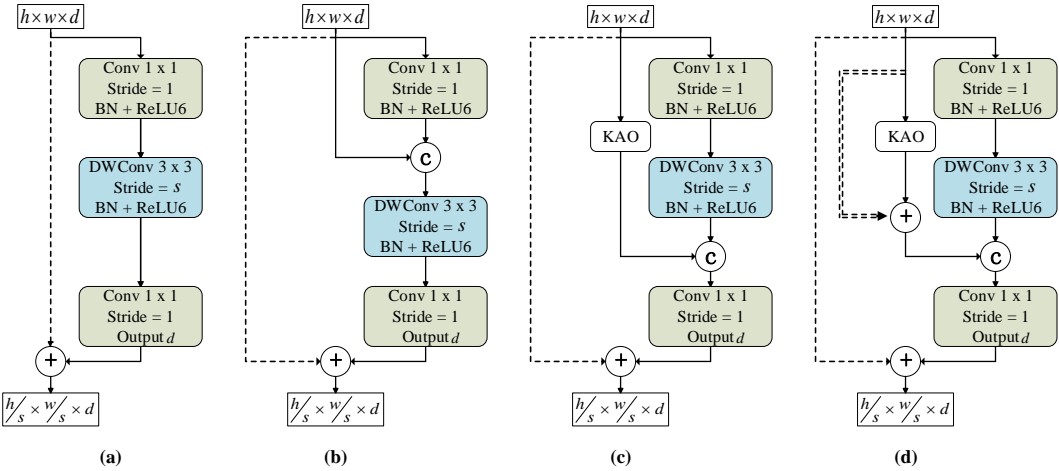

Figure 3: Architectures of the BaseModule (a), BaseSkipModule (b), AttnModule (c), and AttnSkip-Module (d) as described in Section 3.2. The skip connections indicated by single dashed paths are not used when $s > 1$ or $c \neq d$. Those indicated by double dashed paths are not used when $s > 1$.

## 4 THE KANETS ARCHITECTURE

Table 6 describes the detailed KANets architecture. We use KAOs in every AttnSkipModule. In KAOs, we use multi-head attention with 4 heads and concatenate results for output. The linear

Table 6: Details of the KANets architecture. Each line describes a sequence of operators in the format of "input size / operator name / expansion rate $r$ / number of output channels $c$ / number of operators in the sequence $n$ / stride $s$". "Conv2D" denotes the regular 2D convolutional layer. "AvgPool" and "FC" denote the global average pooling layer and the fully-connected layer, respectively. All depth-wise convolutions use the kernel size of $3 \times 3$. For multiple operators in a sequence denoted in the same line, all operators produce $c$ output channels. And the first operator applies the stride of $s$ while the following operators applies the stride of 1. $k$ denotes the class number in the task.

| Input | Operator | $r$ | $c$ | $n$ | $s$ |
|---|---|---|---|---|---|
| $224^2 \times 3$ | Conv2D $3 \times 3$ | - | 32 | 1 | 2 |
| $112^2 \times 32$ | BaseSkipModule | 1 | 16 | 1 | 1 |
| $112^2 \times 16$ | BaseSkipModule | 6 | 24 | 2 | 2 |
| $56^2 \times 24$ | BaseSkipModule | 6 | 32 | 2 | 2 |
| $28^2 \times 32$ | AttnSkipModule | 6 | 32 | 1 | 1 |
| $28^2 \times 32$ | BaseSkipModule | 6 | 64 | 1 | 2 |
| $14^2 \times 64$ | AttnSkipModule | 6 | 64 | 3 | 1 |
| $14^2 \times 64$ | AttnSkipModule | 6 | 96 | 3 | 1 |
| $14^2 \times 96$ | BaseSkipModule | 6 | 160 | 1 | 2 |
| $7^2 \times 160$ | AttnSkipModule | 6 | 160 | 2 | 1 |
| $7^2 \times 160$ | AttnSkipModule | 6 | 320 | 1 | 1 |
| $7^2 \times 320$ | Conv2D $1 \times 1$ | - | 1280 | 1 | 1 |
| $7^2 \times 1280$ | AvgPool + FC | - | $k$ | 1 | - |

transformation is only performed on the value matrix $V$ to limit the number of parameters and computational resources.

## 5    PROOF OF THEOREM 1

The fact that $\mathbf{\Omega}$ and $\mathbf{\Psi}$ are diagonal implies independence in the case of multivariate normal distributions. Therefore, it follows from assumptions (a) and (b) that

$$[\mathbf{r}_1, \mathbf{r}_2, \ldots, \mathbf{r}_h]^T \sim \mathcal{MN}_{h \times w} \left( \mathbf{M}_r, \mathbf{I}_{[h]} \otimes \frac{\overline{\mathbf{\Omega} + \mathbf{\Psi}}}{h} \right), \tag{7}$$

where $\mathbf{M}_r = \overline{\boldsymbol{\mu}} + [\boldsymbol{v}, \boldsymbol{v}, \ldots, \boldsymbol{v}]^T = \overline{\boldsymbol{\mu}} + \mathbf{1}_{[h]} \boldsymbol{v}^T$, and

$$[\mathbf{c}_1, \mathbf{c}_2, \ldots, \mathbf{c}_w] \sim \mathcal{MN}_{h \times w} \left( \mathbf{M}_c, \frac{\overline{\mathbf{\Psi} + \mathbf{\Omega}}}{w} \otimes \mathbf{I}_{[w]} \right), \tag{8}$$

where $\mathbf{M}_c = \overline{\boldsymbol{v}} + [\boldsymbol{\mu}, \boldsymbol{\mu}, \ldots, \boldsymbol{\mu}] = \overline{\boldsymbol{v}} + \boldsymbol{\mu} \mathbf{1}_{[w]}^T$.

Given assumption (c) and $\tilde{\mathbf{X}} = [\mathbf{r}_1, \mathbf{r}_2, \ldots, \mathbf{r}_h]^T + [\mathbf{c}_1, \mathbf{c}_2, \ldots, \mathbf{c}_w]$, we have

$$\tilde{\mathbf{X}} \sim \mathcal{MN}_{h \times w} \left( \tilde{\mathbf{M}}, \frac{\overline{\mathbf{\Psi} + \mathbf{\Omega}}}{w} \oplus \frac{\overline{\mathbf{\Omega} + \mathbf{\Psi}}}{h} \right), \tag{9}$$

where $\tilde{\mathbf{M}} = \mathbf{M}_r + \mathbf{M}_c = (\boldsymbol{\mu} \oplus \boldsymbol{v}) + (\overline{\boldsymbol{\mu}} + \overline{\boldsymbol{v}})$.

If $h = w$, we have

$$\text{tr}(\mathbf{\Omega} \oplus \mathbf{\Psi}) = h \left( \sum \Omega_{ii} + \sum \Psi_{jj} \right), \tag{10}$$

and

$$
\begin{aligned}
& \operatorname{tr}\left(\frac{\overline{\boldsymbol{\Psi}} + \boldsymbol{\Omega}}{w} \oplus \frac{\overline{\boldsymbol{\Omega}} + \boldsymbol{\Psi}}{h}\right) \\
= \quad & \operatorname{tr}\left(\frac{1}{h}(\boldsymbol{\Omega} \oplus \boldsymbol{\Psi}) + \frac{1}{h}(\overline{\boldsymbol{\Psi}} + \overline{\boldsymbol{\Omega}})\right) \\
= \quad & \left(\sum \Omega_{ii} + \sum \Psi_{jj}\right) + h(\overline{\boldsymbol{\Psi}} + \overline{\boldsymbol{\Omega}}) \\
= \quad & 2\left(\sum \Omega_{ii} + \sum \Psi_{jj}\right) \\
= \quad & \frac{2}{h} \cdot \operatorname{tr}\left(\boldsymbol{\Omega} \oplus \boldsymbol{\Psi}\right).
\end{aligned}
\tag{11}
$$

This completes the proof of the theorem.

## REFERENCES

All citations refer to the references in the main paper.

