# OpenReview forum: "Kronecker Attention Networks"
_ICLR.cc/2020/Conference — Reject_

### Official Review · AnonReviewer3 · 2019-10-08
**Official Blind Review #3**

**Rating:** 3

**Review:**

In this paper, the authors propose to reduce the memory and computation complexity of attention networks applied to 2D data (images) by using, in the attention operator, the mean over the rows and columns of images instead of its vectorized version.
The paper is relatively well written, based on intuitive ideas and including clear figures. Several experimental results are reported but, in my opinion, they are not convincing enough.
Below, a list of issues is presented:
-	I think the term “Kronecker” in the name of the model is not well justified or explained. I don’t see a Kronecker product of matrices in the attention operator. Maybe, only in the case of KAO_{QKV} the outer sum could be written as a Kronecker product of some matrices justifying the name. I think this is not a big issue but it is a little bit annoying to use a term that is not naturally introduced to the reader.
-	In my opinion, the main issue is that taking the mean over columns and rows as the only information to construct the attention operator seems not able to capture all relevant information about the underlying dataset. I think there is not a theory that justifies this approach. For example, it could be possible that other linear or nonlinear combination of rows and columns provides better attention operators.
-	It is not clear how the theoretical results (Theorem 1) helps to support the proposed method. Maybe the authors could better explain the implications of their theoretical results. I found the introduction of the method clear but rather disconnected from the theoretical sections 3.3-3.5.
-	Experimental results are not convincing. I cannot agree with the author’s claim that “KANets significantly outperform the previous state-of-the-art compact models”. Below I make comments to each of the presented experimental results.
o	The experimental results about memory and time used by regular attention operator compared with KAOs is obvious and not relevant because their theoretical complexities are well known.
o	In Table 2, it is shown that the proposed method is better than other non-attentional networks. I think this result is not relevant. On the other side, in Table 3, they compare the proposed method against two other attentional networks and show that the proposed method is not better than previous methods in terms of performance. The only advantage seems to be a slight reduction in number of operations which is marginal.
o	In the Ablation Studies (Table 4), the method is not compared with other classical attention network. They only compared against MobileNet. I wonder how the performance results would be if classical attention networks were used having the same level of number of parameters.
o	The results on Image Segmentation are not relevant because the proposed method is not the best one.


**Experience Assessment:**

I have published one or two papers in this area.

**Review Assessment: Checking Correctness Of Derivations And Theory:**

I assessed the sensibility of the derivations and theory.

**Review Assessment: Checking Correctness Of Experiments:**

I carefully checked the experiments.

**Review Assessment: Thoroughness In Paper Reading:**

I read the paper at least twice and used my best judgement in assessing the paper.

---

### Official Review · AnonReviewer2 · 2019-10-23
**Official Blind Review #2**

**Rating:** 3

**Review:**

=== Summary ===
The authors propose the Kronecker Attention Operator (KAO), a novel efficient attention mechanism for high order data such as images (height H and width W dimensions). Regular self-attention on images produce HW queries attend to HW keys and their values, resulting in quadratic time/space complexity O((HW)^2).
In contrast, KAO applies self-attention after averaging keys, values (and potentially queries) along width or height dimension, resulting in (H+W) keys and values and thus achieving linear time/space complexity in O(HW).
The authors analyze their approach from a probabilistic perspective by assuming that the rows and columns follow matrix-variate normal distributions with Kronecker covariance.
Experimental results on ImageNet classification and PASCAL segmentation show that adding Kronecker Attention Operators to baseline architectures yields improvements with a negligible increase in parameters/computations.



=== Recommendation ===
Attention has been shown to provide improvements over convolutions but can indeed incur significant memory costs. Hence, discovering efficient attention mechanisms in an important research direction with practical impact.
The proposed method is sound and simple and may achieve significant speedups over regular self-attention.

However, there are many experimental presentation/evaluation issues in the draft:
- An important assumption of the method is that Kronecker factorization preserves structure well while reducing computations. Previous works reduce memory costs by pooling the input to the attention operator (and resizing accordingly the output). A natural baseline would be to pool in such a way that the number of queries is the same as when applying Kronecker factorization. Can the authors report results on such a baseline?
- Section 4.1 and Table 1 present computational efficiency (memory and latency) in simulated cases. Can the authors additionally report results for the gains obtained on the entire architectures? The rest of the architecture isn't an external factor to ignore.
- Can the authors also report memory consumption and latency time of KANet vs MobileNetv2?
- The authors ignore a vast literature on (self-)attention mechanisms used in vision, some of which are relatively cheap. These methods should also be cited as related work and potentially use as baselines (Squeeze-and-Excitation for example).

The probabilistic analysis does not convincingly motivate the Kronecker Attention Operator. Notably, the assumptions for Theorem 1 are also quite unrealistic for images (nearby pixels are correlated via locality, so rows and columns aren't independent). Can the authors clarify the assumptions and explain why they're reasonable?

In conclusion, the authors only compare their approach against regular self-attention and do not refer to relevant work on attention mechanisms for vision. They do not provide memory/consumption gains on actual architectures, which makes it hard to evaluate the significance of the work. The theoretical motivation for the method is not convincing. I argue for rejection.

=== Additional questions/comments ===
- Section 2.1: Self-attention is better described as Q = W^Q X, K = W^K X, V = W^V X, O = V softmax(K^TQ). As opposed to Q=K=V, O = W^V V softmax((W^KK)^TW^QQ)
- Since the base architecture KANet is MobileNetv2, there is no need to include VGG, AlexNet, SqueezeNet, in Table 2.

**Experience Assessment:**

I have published one or two papers in this area.

**Review Assessment: Checking Correctness Of Derivations And Theory:**

I assessed the sensibility of the derivations and theory.

**Review Assessment: Checking Correctness Of Experiments:**

I carefully checked the experiments.

**Review Assessment: Thoroughness In Paper Reading:**

I read the paper at least twice and used my best judgement in assessing the paper.

---

### Official Review · AnonReviewer1 · 2019-10-24
**Official Blind Review #1**

**Rating:** 3

**Review:**

Akin to previous work introducing tensor regression layers to leverage the tensor/multimodal structure of inputs [Kossaifi et al., 2017], this paper proposes a Kronecker structured attention operator for self attention models acting on tensor data. The proposed operators enforce various constraints on the query key and value matrices of the attention module, where some (or all) of these matrices are obtained as horizontal and lateral averages of the query matrix; in this context, the query matrix is actually the flattening of a 3rd order tensor, whose slices are referred to as "feature maps". A theoretical analysis investigates the properties of imposing a Kronecker structured covariance matrix on the feature maps. The main theorem provides a way to efficiently construct feature maps satisfying this covariance structure. Experiments on image classification and segmentation are given where the proposed approach shows competitive performances while maintaining a number of parameters on par with efficient architectures such as neural nets.

I find this paper difficult to follow. In particular, I find it difficult to see the relevance of the theoretical analysis part. I think I got it in the end, but I believe the analysis could be better motivated/presented and the consequences of Theorem 1 on design choices or implementation could be clarified. Similarly, the description of the architecture in Sections 3.1 and 3.2 is sometimes given without much motivation, letting the reader wonder there are particular reasons for the choices made on the architecture. I acknowledge that this may be because I am not very familiar with attention networks (but I am familiar with the literature on tensorization of neural networks).

I am not very confident in my assessment but I have the feeling that the paper could be greatly improved by some restructuring and giving more insights and motivations.

I have a few questions and comments:
- end of section 2.1: what is the purpose of the seperate linear transformation on each input matrix?
- after Eq. (3) I don't think the notation [cross in diamond] has been introduced.
- the reference (Gupta & Nagar 2018) does not seem appropriate (I don't believe the notion of vectorization requires a reference)
- Some relevant literature may be missing, e.g.,
[A Tensorized Transformer for Language Modeling]
[Tensorized Self-Attention: Efficiently Modeling Pairwise and Global Dependencies Together]


**Experience Assessment:**

I do not know much about this area.

**Review Assessment: Checking Correctness Of Derivations And Theory:**

I assessed the sensibility of the derivations and theory.

**Review Assessment: Checking Correctness Of Experiments:**

I did not assess the experiments.

**Review Assessment: Thoroughness In Paper Reading:**

I made a quick assessment of this paper.

---

### Decision · Program_Chairs · 2019-12-19

**Decision:**

Reject

**Comment:**

This submission has been assessed by three reviewers who scored it as 3/3/3. The main criticism includes lack of motivation for sections 3.1 and 3.2, comparisons to mere regular self-attention without encompassing more works on this topic, a connection between Theorem 1 and the rest of the paper seems missing. Finally, there exists a strong resemblance to another submission by the same authors which is also raises the questions about potentially a dual submission. Even excluding the last argument, lack of responses to reviewers does not help this case. Thus, this paper cannot be accepted by ICLR2020.